# Towards a fuller understanding of neurons with Clustered Compositional Explanations

**Biagio La Rosa**
Sapienza University of Rome
Rome, IT 00185
`larosa@diag.uniroma1.it`

**Leilani H. Gilpin**
University of California, Santa Cruz
Santa Cruz, CA 95060
`lgilpin@ucsc.edu`

**Roberto Capobianco**
Sony AI
Schlieren, CH 8952
`roberto.capobianco@sony.com`

## Abstract

Compositional Explanations [30] is a method for identifying logical formulas of concepts that approximate the neurons' behavior. However, these explanations are linked to the small spectrum of neuron activations (i.e., the highest ones) used to check the alignment, thus lacking completeness. In this paper, we propose a generalization, called *Clustered Compositional Explanations*, that combines Compositional Explanations with clustering and a novel search heuristic to approximate a broader spectrum of the neuron behavior. We define and address the problems connected to the application of these methods to multiple ranges of activations, analyze the insights retrievable by using our algorithm, and propose desiderata qualities that can be used to study the explanations returned by different algorithms.

## 1 Introduction

EXplainable AI (XAI) promises to foster trust [40, 13] and understanding in AI systems. This is particularly important for deep learning (DL) models, which are not understandable. By using XAI methods, we may be able to interpret and better understand the underlying DL processes [24, 23]. Indeed, it is still unclear what kind of knowledge these models learn and how they are able to achieve high performance across many different tasks.

This paper focuses on the methods that explain what neurons learn during the training process. Most of these approaches adopt the idea of investigating what kind of information overstimulates a neuron. For example, generative approaches iteratively change the input features to maximize the neuron's activation [37, 26, 11] or dataset-based approaches select samples where the neuron activates the most [2, 3, 30]. Among them, we are interested in methods that use concepts to explain neurons' behavior. The seminal work of the area is Network Dissection (NetDissect) [2]. NetDissect proposes to use a dataset annotated with concepts to probe the neuron and select, as an explanation, the concept whose annotations are aligned (i.e., overlap) the most with the highest activations. While initially proposed for quantifying the latent space's interpretability, NetDissect has been extensively used for describing and understanding the concepts recognized by neurons [21, 32].

Compositional Explanations [30] (CoEx) generalize NetDissect by extracting logical formulas in place of single concepts, using a beam search over the formulas space. While in principle, these algorithms could be applied to any activation, the current literature [2, 30, 19, 28] focuses only on the exceptionally high activations (i.e., activations above the 0.005 percentile). We argue that using an

37th Conference on Neural Information Processing Systems (NeurIPS 2023).

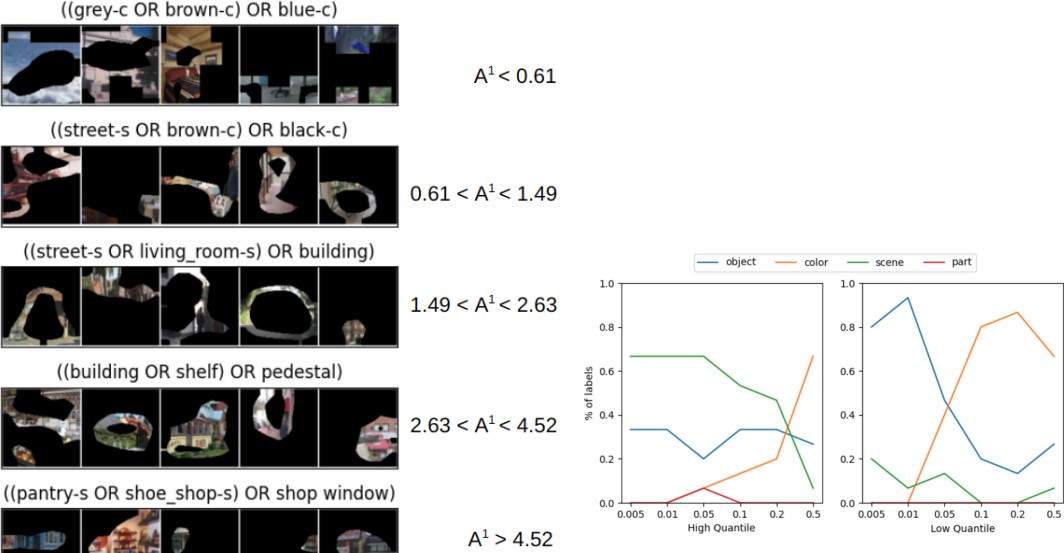

Figure 1: Compositionality captured at different ranges (right side) by unit #1. The left side includes five randomly extracted images inside the range and the labels assigned by our algorithm.

Figure 2: Threshold impact on the category of the returned labels. We can observe that lowering the threshold penalizes some label categories and rewards others. Moreover, given the same threshold, the results change if we apply the threshold starting from the highest activations (left) or lowest activations (right).

arbitrarily high value for the threshold gives us only a partial view of the concepts recognized by a neuron. For example, when these algorithms are used to compare the interpretability of latent space, they can flag as uninterpretable a latent space that is interpretable at lower activations, or vice-versa. When used for downstream tasks [42, 7, 3, 25] the results of these techniques can easily mislead users, mining their trust into explainability [14]. Moreover, in preliminary experiments, we observe that the network uses multiple ranges of activations in the decision process (Appendix E) and that threshold variations lead to different explanations and that the lowest and highest activations react differently to the variation, being associated with different categories of explanations (Figure 2 and Appendix F), thus suggesting that current approaches provide a partial view.

**Contributions**  This paper contributes a generalization of CoEx at different activation ranges. Our insights provide a more detailed description of the neurons behavior. In order to illuminate this problem, we mathematically describe and mitigate the issues connected to the computation of explanations for a broader range of activations. We propose an algorithm (*Clustered Compositional Explanations*), heuristic (*MMESH*), and qualities ready to be used to study and analyze the returned explanations. In detail, we: (i) propose a faster algorithm to compute compositional explanations based on a heuristic; (ii) provide an analysis of a wider spectrum of activations by clustering them and applying our algorithm to the clusters; (iii) extract and discuss the novel insights on image data retrieved from our analysis, namely the presence of *unspecialized activations* and the phenomenon of *progressive specialization*; (iv) collect and identify desirable qualities for this kind of explanation; we also propose three novel ones: sample coverage, activation coverage, and label masking. The code is available at `https://github.com/KRLGroup/Clustered-Compositional-Explanations`.

The manuscript is organized as follows: Section 2 describes related work; Section 3 describes our proposed generalization and desiderata properties; Section 4 analyzes and discusses the proposed generalization and insights retrievable by it. The code will be released upon acceptance.

## 2  Related Work

According to the recent surveys on the topic of explaining individual neurons [6, 39, 15], we can distinguish between feature synthesis and dataset-based approaches. The former aims at generating

inputs that maximally (or minimally) activate a neuron [37, 26, 11] either using DNNs [33, 35] or iterative algorithms [11, 37]. Despite their popularity, these methods face several challenges: the abstraction of the output; they can encode few patterns [34, 37]; the process is stochastic and thus one can get two different visualizations for the same neuron [29]. These problems are addressed by dataset-based approaches, which take a different route and are more effective in helping users [5].

Dataset-based approaches select samples from a dataset where the activation of a neuron is above a given threshold [6]. The advantage is that, by simply looking at the selected samples, one can infer the multiple features detected by the neuron, avoiding the problem of diversity and abstraction. However, the samples-selection can lead to different problems. For example, it becomes difficult to distinguish between causal and merely correlated factors (e.g., is the neuron recognizing only planes or planes into the sky?), and the returned explanations depend on the used dataset [38].

The first problem can be mitigated by looking at concepts instead of entire samples. A concept is a set of semantically related features annotated in a sample. In this case, one can probe for a specific concept to find the neurons responsible for detecting it [8, 10, 18, 1, 31] or fix the neuron and select the concepts where the neuron returns the highest activations [16, 2, 4, 30]. Our work can be placed in dataset-based approaches of the second category. Among them, the work of Bau et al. [2, 4] proposes to select the concept whose annotations are the most aligned to the neuron's activations. To capture more facets of the same activations, Mu and Andreas [30] propose to select logical formulas of concepts instead of a single one using a beam search that extracts the best-aligned formula. The process to extract the logical formulas can be modified to include more operators [16], ontologies [28], or to impose constraints [16, 27]. Our work generalizes both of these works by associating logical formulas with multiple activation ranges per neuron. With respect to these works, our proposed method uses a heuristic search in place of an exhaustive search, uses clustering to compute thresholds instead of using a fixed ad-hoc threshold, and considers multiple intervals of activations and thus the full spectrum of the neuron's activations.

Finally, the problem of dataset dependency has been the focus of recent work, which propose to get natural language explanations by using captioning methods [19] or multimodal models [36]. With respect to this set of approaches, our paper can be considered orthogonal, since the clustering mechanism and the analysis of a wider range of activation can be used as a starting point for these works. We leave the investigation of this direction for future research.

Most of the works described in this section use the intersection over union score [2, 4, 30, 27] and its variants [28, 16] to quantitatively test the quality of the returned explanations. The idea is that if a method discovers a different set of explanations with a higher score, then it discovers more effective directions on the interpretability of the unit. Other metrics used in literature are usually tailored to properties of the proposed methods ( [36]), the goal of the extension ([16]) or they are used to verify properties of the models [30]). To contribute to the analysis of these explanations, we collect two quantitative metrics from the literature, and we propose three additional ones that are general enough to be applied to any dataset-based approach similar to NetDissect and CoEx. While each of them gives us some information, we argue that only by looking at all of them simultaneously one can have a clear view of the difference between explanations returned by different approaches.

## 3 Algorithm

This section describes our proposed Clustered Compositional Explanations and the heuristic used for making the algorithm practically feasible. Additionally, it describes the desiderata properties of the output of these algorithms. Below, we introduce the terminology used in the rest of the paper:

- $\mathfrak{D}$: a dataset annotated with concepts $C$;
- $\mathfrak{L}^l$: the set of logical connections $L$ of arity $l$ that can be built between concepts of $\mathfrak{D}$;
- $L_\leftarrow \in \mathfrak{L}^{l-1}$ and $L_\rightarrow \in \mathfrak{L}^1$ denotes the left side and the right side of a label of arity $i$ obtained by adding an atomic term to the label at each step, respectively;
- $S(x, L)$: a function that returns the binary mask of the input $x$ for the label $L$;
- $A^k(x)$: the activation map of the $k$ neuron when the input is the sample $x$; [1]

---

[1]We assume $A^k(x)$ is already scaled to the same dimensions of $S(x, L)$ (e.g., by padding or interpolation).

- $M_{[\tau_1,\tau_2]}(x)$: the function that returns a binary mask of the activation $A^k(x)$, setting to 0 all the values smaller than $\tau_1$ or greater than $\tau_2$;

- $n_s$: the maximum size of the segmentation;

- $\theta(x, L)$: a function that masks the input $x$ by keeping only the features connected to the label $L$ visible;

- $IMS_{[\tau_1,\tau_2]}(x, L)$: the intersection size between the label mask $S(x, L)$ and the neuron's activation mask $M_{[\tau_1,\tau_2]}(x)$ computed over the activation range $(\tau_1, \tau_2)$.

Moreover, we represent a binary mask as the set of the indices of its elements equal to 1. Thus $\mathbb{1}M_{[\tau_1,\tau_2]}(x, L)$ can be represented by the cardinality $|M_{[\tau_1,\tau_2]}(x, L)|$.

## 3.1 Clustered Compositional Explanations

Clustered Compositional Explanations generalize CoEx [30] and NetDissect [2] by computing explanations on multiple intervals of activation thresholds. We begin the description by recalling the objective of NetDissect [2] and CoEx [30].

**NetDissect [2]** extracts the single concept whose masks overlap the most with the binary activation mask. Network Dissection uses a single threshold ($\tau^{top}$) to compute the activation mask. Conventionally, $\tau^{top}$ is set to the top 0.005 quantiles for each unit k, i.e., $\tau^{top}$ is determined such that $P(a_i \geq \tau^{top}) = 0.005$, $\forall a_i \in A^k(\mathfrak{D})$. Therefore, the objective can be written as:

$$C^{best} = \underset{C \in \mathfrak{L}^1}{\arg\max}\, IoU(\mathfrak{L}^1, \tau^{top}, \infty, \mathfrak{D}) \tag{1}$$

NetDissect proposes an exhaustive search over the full search space of all the concepts in order to select the best explanation.

**CoEx Algorithm [30]** generalizes the NetDissect algorithm by considering logical formulas of arity $n$ in place of single concepts. Therefore, the objective is changed to:

$$L^{best} = \underset{L \in \mathfrak{L}^n}{\arg\max}\, IoU(\mathfrak{L}^n, \tau^{top}, \infty, \mathfrak{D}) \tag{2}$$

When the number of concepts or the dataset size is large enough, the computation of an exhaustive search becomes prohibitive. To solve the issue, Mu et al. [30] propose to use a beam search of size $b$. At each step $i$, only the $b$ best labels of arity $i$ are used as bases for computing labels of arity $i + 1$. The first beam is selected among the labels associated with the best scores computed by NetDissect. Ignoring for simplicity the time needed to compute the masks for the labels, it is clear that the CoEx algorithm needs at least $(n-1) \times b$ times the time needed for running NetDissect.

**Clustered Compositional Explanations** generalizes CoEx by returning a set of logical formulas by identifying $n_{cls}$ clusters into the activations of the $k$ neuron and computing the logical formula that better describes each cluster (i.e., the one that maximizes the IoU score) (Figure 1). Specifically, the algorithm finds the solution to the objective

$$L^{best} = \{\underset{L \in \mathfrak{L}^n}{\arg\max}\, IoU(L, \tau_i, \tau_j, \mathfrak{D}), \forall\, [\tau_i, \tau_j] \in \mathrm{T}\} \tag{3}$$

where

$$IoU(L, \tau_1, \tau_2, \mathfrak{D}) = \frac{\sum_{x \in \mathfrak{D}} |M_{[\tau_1,\tau_2]}(x) \cap S(x, L)|}{\sum_{x \in \mathfrak{D}} |M_{[\tau_1,\tau_2]}(x) \cup S(x, L)|} \tag{4}$$

and

$$\mathrm{T} = \{[min(Cls), max(Cls)], \forall Cls \in Clustering(A^k(\mathfrak{D}))\} \tag{5}$$

$Clustering(A^k(\mathfrak{D}))$ returns $n_{cls}$ disjoint clusters of the non-zero activations of the $k$ neuron over the whole dataset, and $T$ is the set of thresholds computed by selecting the minimum and maximum

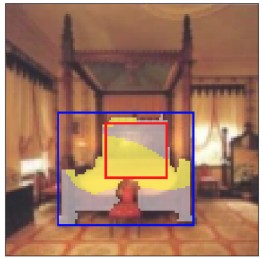

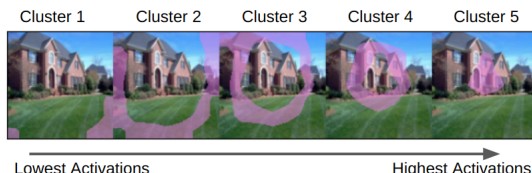

Cluster 1  Cluster 2  Cluster 3  Cluster 4  Cluster 5

Lowest Activations          Highest Activations

Figure 3: Visualization of information used by MMESH: the size of the concept mask (white), size of the intersection (yellow), minimum extension (red bounding box), maximum extension (blue bounding box).

Figure 4: Visualization of activations associated with different clusters. Activations are visualized from the lowest (left, Cluster 1) to the highest ones (right, Cluster 5).

activation inside each cluster. When setting $T = [\tau^{top}, \infty]$, the objective is equivalent to the CoEx algorithm (eq. (2)), while by setting $T = [\tau^{top}, \infty]$ and $L \in \mathfrak{L}^n$, one can obtain the NetDissect objective (eq. (1)). The CoEx algorithm extracts compositional explanations by iteratively applying NetDissect to a beam search tree of width $b$ and deepness $(n-1)$. Thus, since the algorithm applies CoEx on $n_{cls}$ clusters, the vanilla implementation would require $n_{cls} \times (n-1) \times b$ times the computation time of NetDissect. Even employing a parallelization of the computation over units, the base required time and some practical problems[2] arising from the wider considered ranges make the application of the CoEx algorithm practically unfeasible when dealing with multiple clusters.

**Min-Max Extension per Sample Heuristic**    To solve this problem, we propose a beam search guided by a novel, admissible heuristic. Specifically, we propose the *Min-Max Extension per Sample Heuristic* (**MMESH**). Given a sample $x$, a neuron $k$, and a label $L \in \mathfrak{L}^i$, the heuristic estimates the IoU score by combining the following information: the size of the label mask on the sample $S(x, L)$; the coordinates to identify the smallest possible extension of the concept on the sample (i.e., the longest contiguous segment including only concept elements); the coordinates to identify the largest possible extension of the concept on the sample $maxExt(x, L)$; the size of the intersection between the label's terms mask and the neuron's activation on the sample $IMS(x, t) \forall t \in L$.

The first three pieces of information are computed and collected directly from the concept dataset, while the intersection size is collected during the execution of the first step of CoEx for all $L \in \mathfrak{L}^1$ and before expanding the best labels in the current beam for all their terms $t \in \mathfrak{L}^{i-1}$. Note that the shape of the coordinates depends on the data type of the concept dataset. In our case (image data), $minExt(x, L)$ and $maxExt(x, L)$ correspond to the top left and bottom right corners of the largest inscribed rectangle inside the polygon and the largest rectangle patch applied to cover the polygon drawn from the concept's segmentation (i.e., *bounding box*), respectively (Figure 3). MMESH provides an estimation for logical formulas connected by OR, AND, and AND NOT operators and their specialization [16], which are the most used operators [30, 27, 28, 16].

Specifically, the heuristic uses the best-case scenarios for the intersection and the worst-case scenario enhanced by the coordinates for the label's mask to estimate the IoU score. In formulas:

$$IoU(L, \tau_1, \tau_2, \mathfrak{D}) = \frac{\widehat{I}}{\widehat{U}} = \frac{\sum_{x \in \mathfrak{D}} \widehat{I_x}}{\sum_{x \in \mathfrak{D}} \widehat{U_x}} =$$

$$= \frac{\widehat{I_x}}{\sum_{x \in \mathfrak{D}} |M_{[\tau_1, \tau_2]}(x)| + \sum_{x \in \mathfrak{D}} |\widehat{S(x, L)}| - \widehat{I_x}} \qquad (6)$$

where

$$\widehat{I_x} = \begin{cases} min(|IMS_{[\tau_1, \tau_2]}(x, L_\leftarrow)| + |IMS_{[\tau_1, \tau_2]}(x, L_\rightarrow)|, |M(x)|) & op = OR \quad (7) \\ min(|IMS_{[\tau_1, \tau_2]}(x, L_\leftarrow)|, |IMS_{[\tau_1, \tau_2]}(x, L_\rightarrow)|) & op = AND \quad (8) \\ min(|IMS_{[\tau_1, \tau_2]}(x, L_\leftarrow)|, |M_{[\tau_1, \tau_2]}(x)| - |IMS_{[\tau_1, \tau_2]}(x, L_\rightarrow)|) & op = AND\ NOT \quad (9) \end{cases}$$

---

[2]A greater number of zeros in the matrix allows a faster computation since zeroed rows can be skipped.

and

$$\widehat{S(x,L)} = \begin{cases} max(|S(x,L_\leftarrow)|, |S(x,L_\rightarrow)|, \widehat{S(x, L_\leftarrow \cup L_\rightarrow)}), & op = OR & (10) \\ max(MinOver(L), I_x) & op = AND & (11) \\ max(|S(x,L_\leftarrow)| - MaxOver(L), I_x) & op = AND\ NOT & (12) \end{cases}$$

In the previous formulas, $op$ is the logical connector of the formula, $L_\leftarrow \in \mathfrak{L}^{i-1}$ denotes one of the best labels retrieved in the current beam, $L_\rightarrow \in \mathfrak{L}^1$ denotes the candidate term to be connected through $op$ to the label as the right side, $MaxOver(L)$ is a function that returns the maximum possible overlap between the largest segments marked by the coordinates $maxExt(x, L_\leftarrow)$ and $maxExt(x, L_\rightarrow)$, $MinOver(L)$ is a function that returns the minimum possible overlap between the smallest segments marked by the coordinates $minExt(x, L_\leftarrow)$ and $minExt(x, L_\rightarrow)$, and
$\widehat{S(x, L_\leftarrow \cup L_\rightarrow)} = |S(x,L_\rightarrow)| + |S(x,L_\rightarrow)| - MaxOver(L)$.

Since $\widehat{I}_x$ must be an overestimation of $I$, eq. (7) corresponds to the best-case scenario for OR labels (i.e., disjoint masks) and eq. (8) and eq. (9) correspond to the best-case scenario for fully overlapping masks in the case of AND and AND NOT labels. Conversely, since $\widehat{S(x,L)}$ must underestimate the real label's mask and thus cover the worst-case scenario, it assumes fully overlapping maps for OR labels and disjoint maps for AND and AND NOT operators. Note that in the case of AND and AND NOT, the coordinates for the label's mask (i.e., minimum possible overlapping between polygons generated by the coordinates) help us to avoid setting it $\widehat{S(x,L)}$ to 0. We prove that this heuristic is admissible (Appendix A.1); thus, the heuristic search is guaranteed to find the optimal formula inside the beam.

## 3.2 Desiderata Qualities

This section describes a set of statistics and metrics that can be used to describe the qualities of the returned explanations. As mentioned in the previous sections, compositional explanations are commonly analyzed by looking at their IoU score. However, IoU can be artificially increased by increasing the formula's length [30, 27]. Conversely, we promote the simultaneous usage of a set of metrics to have the full view of the efficacy of a method, since each of them has its weak spot when taken in isolation. Additional and alternative statistics are listed in Appendix D.

**Intersection Over Union** The metric optimized by the approaches considered in this paper. It measures the alignment between labels' annotations and activation maps (eq. (4)). Given the activation range $(\tau_1, \tau_2)$, a higher IoU means the algorithm can better capture the pre-existent alignment [30].

**Detection Accuracy** The percentage of masks of the label $L$ overlapping with the neuron activation inside the activation range $(\tau_1, \tau_2)$ [27]. A value closer to one means that most of the label's masks are usually recognized by the neuron in that activation range.

$$DetAcc(L, \tau_1, \tau_2, \mathfrak{D}) = \frac{\sum_{x \in \mathfrak{D}} |M_{[\tau_1, \tau_2]}(x) \cap S(x,L)|}{\sum_{x \in \mathfrak{D}} |(S(x,L)|} \tag{13}$$

**Samples Coverage** The percentage of samples satisfying the label where the neuron activation is inside the activation range $(\tau_1, \tau_2)$. A value closer to one means that the neuron usually recognizes the concept using that activation range.

$$SampleCov(L, \tau_1, \tau_2, \mathfrak{D}) = \frac{|\{x \in \mathfrak{D} : |M_{[\tau_1, \tau_2]}(x) \cap S(x,L)| > 0\}|}{|\{x \in \mathfrak{D} : |(S(x,L)| > 0\}|} \tag{14}$$

**Activation Coverage** The percentage of neuron activations inside the activation range $(\tau_1, \tau_2)$ overlapping with the label's annotations. A value closer to one means that the label captures most of the behavior of this type of activation (i.e., there is a strong mapping).

$$ActCov(L, \tau_1, \tau_2, \mathfrak{D}) = \frac{\sum_{x \in \mathfrak{D}} |M_{[\tau_1, \tau_2]}(x) \cap S(x,L)|}{\sum_{x \in \mathfrak{D}} |M_{[\tau_1, \tau_2]}(x)|} \tag{15}$$

**Explanation Coverage** The percentage of dataset samples covered by the explanations, i.e., samples that satisfy the explanation's label and where the neuron fires inside the activation range $(\tau_1, \tau_2)$. A value closer to one means that the neuron fires at the given activation range when the input includes the explanation's label. Thus, there is a strong correlation between the label and the activation range.

$$ExplCov(L, \tau_1, \tau_2, \mathfrak{D}) = \frac{|\{x \in \mathfrak{D} : |M_{[\tau_1, \tau_2]}(x) \cap S(x, L)| > 0\}|}{|\{x \in \mathfrak{D} : |M_{[\tau_1, \tau_2]}(x)| > 0\}|} \quad (16)$$

**Label Masking** The cosine similarity computed by comparing the neuron's activations when the model is fed by using the full input $x$ and the masked input $\theta(x, L)$. A high score indicates a strong connection between the label and the activation range. Note that we only keep track of the changes in the regions identified by the activation range $\tau_1, \tau_2$.

$$LabMask(L, \tau_1, \tau_2, \mathfrak{D}) = \frac{\sum_{x \in \mathfrak{D}} CosineSim(M^k_{[\tau_1, \tau_2]}(x) A^k(\theta(x, L)), M^k_{[\tau_1, \tau_2]}(x) A^k(x))}{|\{x \in \mathfrak{D} : |M_{[\tau_1, \tau_2]}(x)| > 0\}|} \quad (17)$$

## 4 Analysis

### 4.1 Setup

For the experiments in this section, we follow the same setup of Mu and Andreas [30] with the addition of the set of thresholds used to compute the activation ranges. We fix the length of the explanation to 3, as commonly done in the current literature [28, 27]. For space reasons, in almost all the experiments in this section, we report the results using the last layer of ResNet18 [17] as a base model and Ade20k [44, 46] as a concept dataset. However, the claims hold across different architectures and concept datasets, as reported in Appendix B. We use K-Means as a clustering algorithm and fix the number of clusters to five. The choice of K-Means is motivated by the need for a fast clustering algorithm that aggregates activations associated with a shared semantic and that can be applied to a large quantity of data (see Appendix G for further details about the choice of the clustering algorithm). The number has been set to five because a higher number of clusters returns almost the same labels but repeated over more clusters, and, on average, the scores are lower (Appendix C).

### 4.2 Heuristic Effectiveness

This section compares the number of labels for which the algorithm computes the true IoU (*visited states*) needed by the vanilla CoEx algorithm, our heuristic, and alternative heuristics.

We select and test three heuristics using different amounts of the needed information to estimate the IoU score: the vanilla CoEx algorithm where no heuristics are used (*CoEx*); a heuristic that uses only the size of the label masks per sample (*Areas*); the *Coordinates-Free Heuristic* (*CFH*), an ablated version of our proposed heuristic that does not estimate the size of the label's mask; and our proposed heuristic (*MMESH*). Refer to Appendix A.2 for further details about the baselines.

Table 1 compares the average number of states visited during the computation of the baselines and our MMESH. The results are computed over 100 randomly selected units fixing $T = [\tau^{top}, \infty]$ as

Table 1: Avg. and standard deviation of visited states per unit. Results are computed for 100 randomly extracted units.

| Heuristic | Info from | | Visited States |
|---|---|---|---|
| | NetDissect | Dataset | |
| CoEx | - | - | $39656 \pm 12659$ |
| + Areas | - | ✓ | $23602 \pm 3420$ |
| + CFH | ✓ | ✓ | $5990 \pm 3066$ |
| + MMESH | ✓ | ✓ | $129 \pm 712$ |

in the CoEx algorithm. We can observe that it is possible to lower the number of visited states by increasing the amount of information. The areas heuristic lowers the number of operations by a third and, since the heuristic uses only information from the dataset, it represents a valid option for making NetDissect faster. By adding the estimation of the intersection, we further reduce the number of visited states by one order of magnitude. Finally, *MMESH*, which adds the estimation of the label's mask, reaches the best result, reducing the number of visited states by two orders of magnitude.

Practically, this result means that, for each unit, we can generate explanations for clusters in the same amount of time (or less) as the vanilla compositional algorithm as long as the number of clusters is reasonably low. Indeed, while the CoEx algorithm takes, on average, more than ~60 minutes per unit, our proposed *MMESH* takes less than 2 minutes. [3]

## 4.3 Explanations Analysis

In this section, we analyze the explanations produced by NetDissect, CoEx, and our algorithm in terms of the desiderata qualities introduced in Section 3.2.

Table 2 shows that NetDissect and CoEx reach similar values in most considered scores, and the labels returned by CoEx are only slightly better than the NetDissect ones. The most significant margin can be observed in Sample Coverage, Detection Accuracy, and IoU. The higher IoU can be explained by the degenerate effect of increasing the formula's length [30]. The margin in Sample Coverage and Detection Accuracy means that the neuron fires in most of the samples annotated with labels returned by NetDissect. However, since the Detection Accuracy is lower, the overlap between activations and annotation is less consistent and more sparse. This is verified by the observation that most of the CoEx labels are connected by OR operators, thus increasing the number of candidate samples, and that most of the NetDissect labels are scene concepts (i.e., concepts that cover the full input), which are associated with a sparse overlapping due to the size of activation ranges. Looking at the average over the clusters reached by Clustered Compositional Explanations, the algorithm seems to return better labels with respect to almost all the desiderata qualities by a large margin. However, if we look at the average scores per cluster, we can note that the average is favored by Cluster 1 and Cluster 2. In Table 2, clusters are named progressively according to the activation ranges. Thus, Cluster 1 corresponds to the lowest activations and Cluster 5 to the highest activations.

First, we can note that Cluster 1 and 2 have an almost perfect Dataset and Sample Coverage. By the definition of the scores, this means that their labels cover the full dataset and that there is a strong connection between the activation range and the label. These extreme values motivated us to further investigate these clusters, which are discussed in the next section. Now, we can analyze Cluster 5, which includes the highest activations and, thus, also the range used by NetDissect and CoEx. We observe that with respect to CoEx, enlarging the range of activation has a marginal impact on Detection Accuracy and Label Masking, no effect on IoU and Sample Coverage, and a positive effect on Explanation Coverage and Activation Coverage. Combining the scores, we can infer that the larger activations (higher Activation Coverage) allow the algorithm to detect labels associated with slightly bigger concepts and better aligned to the neuron behavior (the same IoU and a higher Explanation Coverage). Finally, Cluster 3 and Cluster 4, which include intermediate activations,

---

[3]Timing collected using a workstation powered by an NVIDIA GeForce RTX-3090 graphic card.

Table 2: Avg. and Std Dev. of the desiderata qualities over the labels returned by NetDissect, CoEx, and Clustered Compositional Explanations (our).

|  | IoU | ExplCov | SampleCov | ActCov | DetAcc | LabMask |
|---|---|---|---|---|---|---|
| NetDissect | $0.05 \pm 0.03$ | $0.14 \pm 0.20$ | $0.58 \pm 0.27$ | $0.14 \pm 0.17$ | $0.15 \pm 0.15$ | $0.62 \pm 0.26$ |
| CoEx | $0.08 \pm 0.03$ | $0.14 \pm 0.15$ | $0.53 \pm 0.25$ | $0.15 \pm 0.10$ | $0.18 \pm 0.10$ | $0.57 \pm 0.22$ |
| Our | $0.15 \pm 0.07$ | $0.60 \pm 0.36$ | $0.69 \pm 0.24$ | $0.37 \pm 0.17$ | $0.20 \pm 0.09$ | $0.61 \pm 0.16$ |
| Cluster 1 | $0.27 \pm 0.02$ | $0.99 \pm 0.00$ | $0.96 \pm 0.02$ | $0.54 \pm 0.02$ | $0.34 \pm 0.03$ | $0.64 \pm 0.06$ |
| Cluster 2 | $0.15 \pm 0.02$ | $0.91 \pm 0.16$ | $0.80 \pm 0.06$ | $0.49 \pm 0.09$ | $0.18 \pm 0.03$ | $0.68 \pm 0.07$ |
| Cluster 3 | $0.12 \pm 0.03$ | $0.56 \pm 0.27$ | $0.64 \pm 0.15$ | $0.33 \pm 0.10$ | $0.16 \pm 0.05$ | $0.62 \pm 0.14$ |
| Cluster 4 | $0.10 \pm 0.04$ | $0.35 \pm 0.19$ | $0.52 \pm 0.22$ | $0.25 \pm 0.12$ | $0.15 \pm 0.07$ | $0.53 \pm 0.19$ |
| Cluster 5 | $0.09 \pm 0.05$ | $0.21 \pm 0.18$ | $0.53 \pm 0.26$ | $0.22 \pm 0.15$ | $0.18 \pm 0.09$ | $0.58 \pm 0.23$ |

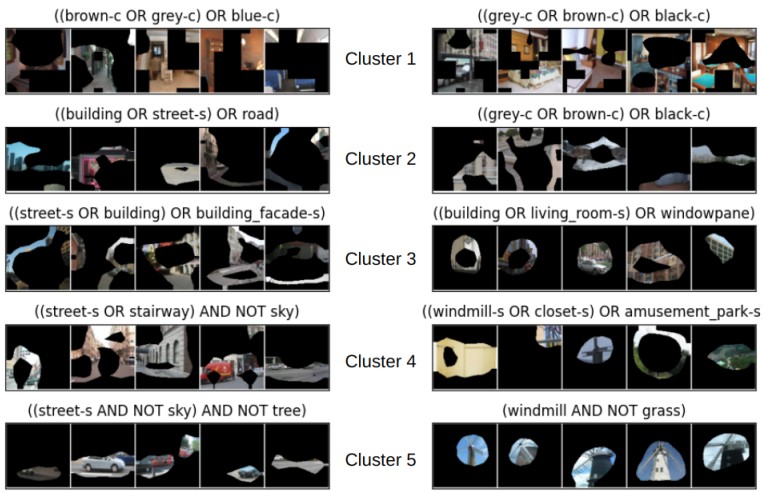

Figure 5: Examples of specialization (left) and polysemy (right). Neuron #455 (left) recognizes streets in more and more specific contexts through the clusters. Neuron #368 (right) recognizes unrelated concepts among different activation ranges (windmill, closet, amusement park).

behave similarly to Cluster 5, but progressively improve the Sample Coverage, IoU, and Explanation Coverage. Therefore, the insight extracted for Cluster 5 holds also for them and, as we discuss in the next section, it is connected to the property of specialization.

In summary, extracting compositional explanations at different and wider activation ranges maintains or improves the same qualities of the returned explanations. Additionally, the combination and analysis of different qualities simultaneously allow us to extract a bigger view of the compositionality of neuron activations, providing hints ready to be exploited by further analyses.

## 4.4 Neurons Compositionality

In this section, starting from the results reported in the previous section, we analyze the compositionality of neurons at a wider range of activation.

**Unspecialized Activations.** As previously discussed, Cluster 1 and Cluster 2 are associated with a high Sample Coverage, meaning that they almost cover the full dataset. Therefore, the labels should be present in almost all the samples. By inspecting the labels, we observe that they are often a union of colors (i.e., Black OR Blue OR Grey) or a mix of colors and general concepts (i.e., Black OR Blue OR Sky), and few labels are repeated over the whole range of neurons. While the first observation can suggest that they recognize general concepts or colors, the second one goes in the opposite direction. To investigate the phenomenon, we applied our algorithm on untrained networks, finding that all the clusters are associated with these kinds of labels in this case. Thus, these labels represent the *default labels* to which the algorithm converges when the activations are random. We call the activations ranges associated with such labels *unspecialized*, meaning that neurons do not recognize specific concepts or purposely ignore the concepts covered by the activation. By analyzing the clusters from one to five, we found (Table 3) that Cluster 1 is almost always associated with unspecialized

Table 3: Percentage of unspecialized and weakly specialized activation ranges in ResNet18 (ReLU Yes) and DenseNet161(ReLU No) over 128 randomly extracted units.

|  | ReLU | Cluster 1 | Cluster 2 | Cluster 3 | Cluster 4 | Cluster 5 |
|---|---|---|---|---|---|---|
| Unspecialized | Yes | 0.93 | 0.37 | 0.01 | 0.00 | 0.00 |
|  | No | 0.05 | 0.60 | 0.95 | 0.30 | 0.05 |
| Weakly Specialized | Yes | 0.00 | 0.03 | 0.01 | 0.01 | 0.00 |
|  | No | 0.05 | 0.12 | 0.02 | 0.17 | 0.02 |

activations and Cluster 2 half of the time. This phenomenon can also be partial, meaning that the first part of the label is assigned to a default label, but the last part converges on a different concept. In this case, we call these activation ranges *weakly specialized*. They are rare, especially in ReLU networks, and usually appear only in the clusters near the unspecialized ones. We hypothesize that they are a side effect of the clustering algorithm, and a future clustering algorithm tailored to extracting the best activation ranges could reduce their number. Table 3 also shows a similar behavior of ReLU and non-ReLU networks when the activations are close to 0. In ReLU layers, activations are stored in Cluster 1, and they are unspecialized 93% of the time. This percentage becomes smaller when we approach the higher clusters. We can observe a similar behavior in the case of the layer without ReLU. In this case, since the activations can assume negative values, the activations close to zero are stored in the middle clusters, and thus, Cluster 3 includes unspecialized activations 95% of the time. And again, when we move far away from zero, the percentage starts to decrease, as in the ReLU layers.

**Progressive Specialization.** Progressive specialization is a phenomenon we observe in association with ReLU layers, where lower activations recognize more general concepts (e.g., building, sky, etc.), and higher ones progressively detect more complex objects. The phenomenon is similar to the one observed by Zhou et al. [43], in which the lower layers of a neural network detect more abstract concepts while the latest detect the most complex and specific ones. In the case of image data, this phenomenon seems to be also spatially aligned, meaning that lower activations capture external elements surrounding the objects detected by higher activations (Figure 4). The specialization property highlighted by Mu and Andreas [30] is an example of specialization (Figure 5).

**Activation Polysemy** Following Mu and Andreas [30], we manually inspected 128 randomly extracted units to analyze the relations among concepts inside the returned labels and among activation ranges. A *polysemic* neuron is the one that fires for unrelated concepts [30]. Mu and Andreas [30] found that 31% of neurons are polysemic in the highest activation ranges. We explore the polysemy considering the full range of activations, meaning that a neuron is considered non-polysemic only if all the labels associated with the clusters are related. While, as expected, the number of polysemic neurons is much larger (~85%), it is interesting to note that ~15% of neurons fire for related concepts at any activation range, meaning that they are *highly specialized*.[4]

## 5   Limitations and Future Work

This paper presented a first step towards a fuller understanding of neurons compositionality. We introduced Clustered Compositional Explanation, an algorithm that clusters the activations and then applies the CoEx algorithm guided by heuristics to them. Using our algorithm, we found and analyzed novel phenomena connected to the neuron's activations, such as the unspecialization of the lowest activations in ReLU networks and the progressive specialization. Despite the progress, there are some limitations of the current design. First, the labels returned by our algorithm are deeply connected to the activation ranges identified by the clustering algorithm. Therefore, future work could analyze the differences among different clustering algorithms or develop a novel one tailored to the given task. The extracted insights refer to the image data case. While the heuristic and the approach are domain agnostic, the application on a different domain could extract different kinds of insights, thus limiting or confirming the generality of the findings of Section 4.4. We hypothesized that looking at the scores of each cluster can uncover a deeper understanding of the behavior of different activation ranges. However, an interesting direction could be the development of weighting mechanisms to weight the contribution of each cluster to the final averaged scores, which is desirable when the number of clusters is high and looking and comparing individual cluster can become problematic. Finally, the specific labels returned by these algorithms are linked to the concept dataset used to probe the neurons, as observed by Ramaswamy et al. [38]. While the general insights hold even when changing the dataset (Appendix B.2), we do not address the problem of returning the same specific labels across different datasets. Other than mitigating the above-mentioned limitations, future work could also explore the application of the heuristic to study the optimality of the returned explanations, or the application of clusters on recent methods for reducing the dependency on the concepts datasets [36, 19].

---

[4]Note that the evaluation is subjective. Therefore, the reported numbers must be considered as an indication.

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

# A  Heuristics

## A.1  Proof of Admissibility of MMESH

Since the heuristic aims at approximating the IoU score, we can start by expanding the denominator of the IoU formula:

$$
\begin{aligned}
IoU(L, \tau_1, \tau_2, \mathfrak{D}) &= \frac{I}{U} \\
&= \frac{\sum_{x \in \mathfrak{D}} |M_{[\tau_1, \tau_2]}(x) \cap S(x, L)|}{\sum_{x \in \mathfrak{D}} |M_{[\tau_1, \tau_2]}(x) \cup S(x, L)|} \\
&= \frac{I_x}{\sum_{x \in \mathfrak{D}} |M_{[\tau_1, \tau_2]}(x)| + \sum_{x \in \mathfrak{D}} |S(x, L)| - I}
\end{aligned}
\tag{18}
$$

Specifically, the heuristic should avoid the direct computation of $I$ and $\sum_{x \in \mathfrak{D}} |S(x, L)|$, providing estimations of them. To be admissible, the heuristic cannot underestimate the intersection or overestimate the union. Thus, it must satisfy the following constraints:

$$
\begin{cases}
|\widehat{I_x}| \geq |I_x| & (19) \\
0 \leq |\widehat{S(x, L)}| - |\widehat{I_x}| \leq |S(x, L)| - I & (20)
\end{cases}
$$

$\forall x \in \mathfrak{D}$. Eq. (19) ensures that the heuristic returns an optimistic estimation for the intersection at the numerator. Eq. (20) ensures that the heuristic returns a pessimistic estimation of the denominator of eq. (18).

**Equation (19)**  We begin by proving the first equation. Recall that MMESH estimates $|\widehat{I_x}|$ as:

$$
\widehat{I_x} = \begin{cases}
min(|IMS_{[\tau_1, \tau_2]}(x, L_\leftarrow)| + |IMS_{[\tau_1, \tau_2]}(x, L_\rightarrow)|, |M(x)|) & op = OR & (21) \\
min(|IMS_{[\tau_1, \tau_2]}(x, L_\leftarrow)|, |IMS_{[\tau_1, \tau_2]}(x, L_\rightarrow)|) & op = AND & (22) \\
min(|IMS_{[\tau_1, \tau_2]}(x, L_\leftarrow)|, |M_{[\tau_1, \tau_2]}(x)| - |IMS_{[\tau_1, \tau_2]}(x, L_\rightarrow)|) & op = AND\ NOT & (23)
\end{cases}
$$

Given the masks $L_\leftarrow$ and $L_\rightarrow$, we can distinguish between two cases: overlapping and non-overlapping masks. If the masks do not overlap, then the real intersection is given by:

$$
I_x = \begin{cases}
|IMS_{[\tau_1, \tau_2]}(x, L_\leftarrow)| + |IMS_{[\tau_1, \tau_2]}(x, L_\rightarrow)| & op = OR & (24) \\
0 & op = AND & (25) \\
|IMS_{[\tau_1, \tau_2]}(x, L_\leftarrow)| & op = AND\ NOT & (26)
\end{cases}
$$

Comparing eq.(22-24) and eq. (25-27) we can verify that eq. (19) holds. Indeed, eq. (25) $\leq$ eq. (22) due to the non-negative property of the cardinality and $IMS_{[\tau_1, \tau_2]}(x, L_\rightarrow) \subseteq M_{[\tau_1, \tau_2]}(x)$. eq. (26) $\leq$ eq. (23) can be proved by observing that the mask obtained by $M_{[\tau_1, \tau_2]}(x) - IMS_{[\tau_1, \tau_2]}(x, L_\rightarrow)$ contains $IMS_{[\tau_1, \tau_2]}(x, L_\leftarrow)$ since $L_\leftarrow$ and $L_\rightarrow$ are assumed to be non-overlapping and thus the minimum operator used in the heuristics selects $|IMS_{[\tau_1, \tau_2]}(x, L_\leftarrow)|$. Finally, $IMS_{[\tau_1, \tau_2]}(x, L_\leftarrow) \subseteq M_{[\tau_1, \tau_2]}(x)$ and $IMS_{[\tau_1, \tau_2]}(x, L_\rightarrow) \subseteq M_{[\tau_1, \tau_2]}(x)$ by definition of $IMS_{[\tau_1, \tau_2]}(x, L)$. Since the two masks are not overlapping and $|M_{[\tau_1, \tau_2]}(x)| \geq IMS_{[\tau_1, \tau_2]}(x, L_\leftarrow) + IMS_{[\tau_1, \tau_2]}(x, L_\rightarrow)$ then eq. (24) $\leq$ eq. (21) and thus $I(L) = \widehat{I}(L)$.

Let's begin with fully overlapping masks for the case of overlapping masks. In this case

$$
I_x = \begin{cases}
max(|IMS_{[\tau_1, \tau_2]}(x, L_\leftarrow)|, |IMS_{[\tau_1, \tau_2]}(x, L_\rightarrow)|) & op = OR & (27) \\
min(|IMS_{[\tau_1, \tau_2]}(x, L_\leftarrow)|, |IMS_{[\tau_1, \tau_2]}(x, L_\rightarrow)|) & op = AND & (28) \\
min(|IMS_{[\tau_1, \tau_2]}(x, L_\leftarrow)|, |M_{[\tau_1, \tau_2]}(x)| - |IMS_{[\tau_1, \tau_2]}(x, L_\rightarrow)|) & op = AND\ NOT & (29)
\end{cases}
$$

Comparing eq. (22-24) to eq. (280), it is easy to see that eq. (19) holds for AND and AND NOT operators since eq. (22) = eq. (28) and eq. (23) = eq. (29). When $op = OR$, one can observe that eq. (27) $\leq$ eq. (21) since the maximum between two cardinalities is lower than their sum and $|M_{[\tau_1, \tau_2]}(x)| \geq max(|IMS_{[\tau_1, \tau_2]}(x, L_\leftarrow)|, |IMS_{[\tau_1, \tau_2]}(x, L_\rightarrow)|)$ since the masks are fully overlapping.

The cases described above (fully overlapping and non-overlapping masks) for the estimation of $\widehat{I_x}$ cover the best-case scenarios for all the involved operators. Therefore, when the masks are partially overlapping, the real intersection is smaller than the case of non-overlapping for the OR operator and smaller than the case of fully overlapping for the AND and AND NOT operator. Thus, $|\widehat{I_x}| \geq |I_x|$ holds also for partially overlapping masks.

**Equation (20)** Recall that the heuristic computes $\widehat{S(x, L)}$ as:

$$\widehat{S(x, L)} = \begin{cases} max(|S(x, L_\leftarrow)|, |S(x, L_\rightarrow)|, |\widehat{S(x, L_\leftarrow \cup L_\rightarrow)}|) & op = OR & (30) \\ max(MinOver(L), |\widehat{I_x}|) & op = AND & (31) \\ max(|S(x, L_\leftarrow)| - MaxOver(L), |\widehat{I_x}|) & op = AND\ NOT & (32) \end{cases}$$

where $\widehat{S(x, L_\leftarrow \cup L_\rightarrow)} = |S(x, L_\rightarrow)| + |S(x, L_\rightarrow)| - MaxOver(L)$. We can distinguish again between the case of non-overlapping and fully overlapping masks.
We proceed by proving for each case that (i) $\widehat{S(x, L)} \leq S(x, L)$, (ii) $\widehat{S(x, L)} - \widehat{I_x} \leq S(x, L) - I_x$, and (iii) $\widehat{S(x, L)} - \widehat{I_x} \geq 0$.

In the case of non-overlapping masks, the real joint label is

$$S(x, L) = \begin{cases} |S(x, L_\leftarrow)| + |S(x, L_\rightarrow)| & op = OR & (33) \\ 0 & op = AND & (34) \\ |S(x, L_\leftarrow)| & op = AND\ NOT & (35) \end{cases}$$

Let us begin by comparing eq. (30) and eq. (33) for the OR operator.
When the max operator selects $|S(x, L_\leftarrow)|$ or $|S(x, L_\leftarrow)|$ (i) is verified since $|S(x, L_\leftarrow)| \leq |S(x, L_\leftarrow)| + |S(x, L_\rightarrow)|$ and $|S(x, L_\rightarrow)| \leq |S(x, L_\leftarrow)| + |S(x, L_\rightarrow)|$ due to the non-negativity of the cardinality. When the maximum is $|\widehat{S(x, L_\leftarrow \cup L_\rightarrow)}|$ (i) is verified since $MaxOver(L)$ by definition returns a cardinality (the one of the overlapping between the largest possible extensions), then $MaxOver(L) \geq 0$ and thus $|\widehat{S(x, L_\leftarrow \cup L_\rightarrow)}| \leq |S(x, L_\leftarrow)| + |S(x, L_\rightarrow)|$. (ii) is proved by observing that $|\widehat{I_x}|$ is an overestimation of $I$ and thus $|\widehat{S(x, L)}| - |\widehat{I_x}| \leq |S(x, L)| - |I|$.
(iii) can be proved by showing that $|\widehat{S(x, L)}| \geq |\widehat{I_x}|$. The proof follows by noting that $IMS_{[\tau_1, \tau_2]}(x, L_\leftarrow) \subseteq S(x, L_\leftarrow), IMS_{[\tau_1, \tau_2]}(x, L_\rightarrow) \subseteq S(x, L_\rightarrow)$, and the max operator selects the highest cardinality.
Now, let us examine the case of the AND operator. In this case, (i), (ii), and (iii) are proved by observing that since the masks are not overlapping and $MinOver(L)$ returns the minimum possible overlap, then $MinOver(L) = 0$. Therefore, the max operator in eq. (31) returns $|\widehat{I_x}|$ and, thus $|\widehat{S(x, L)}| - |\widehat{I_x}| = |S(x, L)| - |I_x| = 0$ due to eq. (25). Finally, (i) also holds for the operator AND NOT since $MaxOver(L) \geq 0$ and thus $|S(x, L_\leftarrow)| - MaxOver(L) \leq |S(x, L_\leftarrow)|$ and $|\widehat{I_x}| \leq |S(x, L_\leftarrow)|$ since eq. (9) can be at maximum $|IMS(x, L_\leftarrow)|$ and $|IMS_{[\tau_1, \tau_2]}(x, L_\leftarrow)| \leq S(x, L_\leftarrow)|$. Since $|\widehat{I_x}| \leq |S(x, L_\leftarrow)|$ and $|S(x, L_\leftarrow) \leq |S(x, L_\leftarrow)|$, then (ii) and (iii) are verified.

Now, let us move to the case of fully overlapping masks. In this case, it holds:

$$S(x, L) = \begin{cases} max(|S(x, L_\leftarrow)|, |S(x, L_\rightarrow)|) & op = OR & (36) \\ min(|S(x, L_\leftarrow)|, |S(x, L_\rightarrow)|) & op = AND & (37) \\ max(|S(x, L_\leftarrow)| - |S(x, L_\rightarrow)|, 0) & op = AND\ NOT & (38) \end{cases}$$

By comparing eq. (36) and eq. (30), (i) holds for the OR operator since masks are fully overlapping, and thus $\widehat{S(x, L_\leftarrow \cup L_\rightarrow)}$ is equal to the largest mask between $S(x, L_\leftarrow)$ and $S(x, L_\rightarrow)$. (iii) is verified because $IMS_{[\tau_1, \tau_2]}(x, L_\rightarrow)|) \subseteq S(x, L_\rightarrow)$ and $IMS_{[\tau_1, \tau_2]}(x, L_\leftarrow)|) \subseteq S(x, L_\leftarrow)$. Finally, (iii) is verified due to the overestimation of $\widehat{I_x}$ and thus eq. (20) holds. In the case of the AND operator, the equation is easily verified by observing that $MinOver(L)$ returns a subset of $S(x, L_\leftarrow) \cap S(x, L_\rightarrow)$ and thus $MinOver(L) \leq min(|S(x, L_\leftarrow)|, |S(x, L_\rightarrow)|)$, and (i) holds. (ii) follows from the property of overestimation of $\widehat{I_x}$. (iii) is trivially verified by the max operator used in eq. (38).

The cases described above (fully overlapping and non-overlapping masks) cover the worst-case scenarios for all the involved operators for the estimation of $|\widehat{S(x, L_\leftarrow)}|$. Therefore, when the masks partially overlap, the real label's mask is greater than the case of non-overlapping for the OR operator and bigger than the case of fully overlapping for the AND and AND NOT operator. Thus, since eq. (20) holds for them, then it also holds for partially overlapping masks.

In conclusion, we proved that $|\widehat{I_x}| \geq |I_x|$ and that $0 \leq |\widehat{S(x, L)}| - |\widehat{I_x}| \leq |S(x, L)| - I$, thus, the heuristic is admissible, and it returns the optimal formula inside the beam.

### A.2 Alternative Heuristics

#### A.2.1 Coordinate-Free Heuristic

This heuristic follows the same structure as the MMESH heuristic, but it does not use the coordinates to compute the minimum and maximum possible extension of the label mask. Practically, it avoids the estimation of $\widehat{S(L, x)}$ by setting it to 0.

$$IoU(L, \tau_1, \tau_2, \mathfrak{D}) = \frac{\widehat{I_x}}{\sum_{x \in \mathfrak{D}} |M_{[\tau_1, \tau_2]}(x)| - \widehat{I_x}} \tag{39}$$

where $\widehat{I_x}$ is defined as in MMESH. This heuristic could be used in place of MMESH in contexts where computing the coordinates of the maximum and minimum extension is too costly.

#### A.2.2 Areas Heuristic

This heuristic does not collect additional info during the first step of the CoEx algorithm. Therefore, the IoU is estimated using only the information about the mask size of terms composing the current label.

$$IoU(L, \tau_1, \tau_2, \mathfrak{D}) = \frac{\widehat{I_x}}{\sum_{x \in \mathfrak{D}} |M_{[\tau_1, \tau_2]}(x)| - \widehat{I_x}} \tag{40}$$

where

$$\widehat{I_x} = \begin{cases} min(|S_{[\tau_1, \tau_2]}(x, L_\leftarrow)| + |S_{[\tau_1, \tau_2]}(x, L_\rightarrow)|, |M(x)|) & op = OR \tag{41} \\ min(|S_{[\tau_1, \tau_2]}(x, L_\leftarrow)|, |S_{[\tau_1, \tau_2]}(x, L_\rightarrow)|) & op = AND \tag{42} \\ min(|S_{[\tau_1, \tau_2]}(x, L_\leftarrow)|, size(x) - |S_{[\tau_1, \tau_2]}(x, L_\rightarrow)|) & op = AND\ NOT \tag{43} \end{cases}$$

The *areas heuristic* could also be used to speed up the vanilla NetDissect algorithm when it is used as a standalone algorithm.

## B  Additional Results

This section shows the comparison between NetDissect, CoEx, and Clustered Compositional Explanations on two additional architectures and one more concept dataset.

### B.1  Other Models

Table 4 and Table 5 show the comparison between NetDissect, CoEx, and Clustered Compositional Explanations when the base model is DenseNet [20] and AlexNet [22]. We can easily observe that the analysis carried on for ResNet also holds in these cases.

### B.2  Pascal Dataset

Table 6 compares NetDissect, CoEx, and Clustered Compositional Explanations when the Pascal dataset [12] is used as a probing concept dataset for the algorithms. Note that the dataset includes only colors, objects, and parts in this case. We can see that the differences among algorithms are similar to the ones discussed in Section 4.3, and thus, the insights are valid across different datasets.

Table 4: Avg. and Std Dev. of the desired qualities over the labels returned by NetDissect, CoEx, and Clustered Compositional Explanations (our) applied to AlexNet. Results are computed for 50 randomly extracted units.

| | IoU | ExplCov | SampleCov | ActCov | DetAcc | LabMask |
|---|---|---|---|---|---|---|
| NetDissect | $0.03 \pm 0.02$ | $0.25 \pm 0.34$ | $0.32 \pm 0.21$ | $0.15 \pm 0.19$ | $0.13 \pm 0.10$ | $0.27 \pm 0.21$ |
| CoEx | $0.05 \pm 0.02$ | $0.24 \pm 0.27$ | $0.22 \pm 0.16$ | $0.14 \pm 0.11$ | $0.11 \pm 0.07$ | $0.23 \pm 0.18$ |
| Our | $0.12 \pm 0.07$ | $0.65 \pm 0.34$ | $0.67 \pm 0.28$ | $0.33 \pm 0.18$ | $0.17 \pm 0.08$ | $0.48 \pm 0.17$ |
| Cluster 1 | $0.25 \pm 0.04$ | $0.99 \pm 0.00$ | $0.91 \pm 0.09$ | $0.57 \pm 0.03$ | $0.30 \pm 0.06$ | $0.55 \pm 0.11$ |
| Cluster 2 | $0.12 \pm 0.03$ | $0.88 \pm 0.20$ | $0.76 \pm 0.16$ | $0.43 \pm 0.13$ | $0.15 \pm 0.03$ | $0.60 \pm 0.13$ |
| Cluster 3 | $0.10 \pm 0.03$ | $0.63 \pm 0.25$ | $0.56 \pm 0.16$ | $0.29 \pm 0.09$ | $0.14 \pm 0.04$ | $0.54 \pm 0.13$ |
| Cluster 4 | $0.08 \pm 0.03$ | $0.47 \pm 0.25$ | $0.37 \pm 0.15$ | $0.23 \pm 0.10$ | $0.12 \pm 0.03$ | $0.42 \pm 0.13$ |
| Cluster 5 | $0.06 \pm 0.02$ | $0.27 \pm 0.25$ | $0.23 \pm 0.10$ | $0.14 \pm 0.10$ | $0.13 \pm 0.06$ | $0.27 \pm 0.14$ |

Table 5: Avg. and Std Dev. of the desired qualities over the labels returned by NetDissect, CoEx, and Clustered Compositional Explanations (our) applied to DenseNet161. Results are computed for 50 randomly extracted units.

| | IoU | ExplCov | SampleCov | ActCov | DetAcc | LabMask |
|---|---|---|---|---|---|---|
| NetDissect | $0.05 \pm 0.03$ | $0.05 \pm 0.04$ | $0.52 \pm 0.30$ | $0.07 \pm 0.05$ | $0.15 \pm 0.09$ | $0.40 \pm 0.26$ |
| CoEx | $0.06 \pm 0.03$ | $0.09 \pm 0.10$ | $0.51 \pm 0.27$ | $0.11 \pm 0.06$ | $0.13 \pm 0.07$ | $0.42 \pm 0.25$ |
| Our | $0.20 \pm 0.08$ | $0.77 \pm 0.30$ | $0.83 \pm 0.20$ | $0.43 \pm 0.15$ | $0.27 \pm 0.10$ | $0.52 \pm 0.27$ |
| Cluster 1 | $0.14 \pm 0.05$ | $0.63 \pm 0.27$ | $0.74 \pm 0.12$ | $0.35 \pm 0.13$ | $0.20 \pm 0.07$ | $0.61 \pm 0.27$ |
| Cluster 2 | $0.26 \pm 0.06$ | $0.96 \pm 0.09$ | $0.96 \pm 0.07$ | $0.53 \pm 0.06$ | $0.35 \pm 0.08$ | $0.48 \pm 0.25$ |
| Cluster 3 | $0.26 \pm 0.03$ | $0.99 \pm 0.01$ | $0.98 \pm 0.04$ | $0.55 \pm 0.02$ | $0.33 \pm 0.05$ | $0.38 \pm 0.20$ |
| Cluster 4 | $0.20 \pm 0.07$ | $0.84 \pm 0.23$ | $0.85 \pm 0.21$ | $0.44 \pm 0.13$ | $0.27 \pm 0.10$ | $0.57 \pm 0.27$ |
| Cluster 5 | $0.11 \pm 0.05$ | $0.42 \pm 0.28$ | $0.62 \pm 0.21$ | $0.28 \pm 0.14$ | $0.18 \pm 0.08$ | $0.56 \pm 0.30$ |

Table 6: Avg. and Std Dev. of the desired qualities over the labels returned by NetDissect, CoEx, and Clustered Compositional Explanations (our) on the Pascal Dataset. Results are computed for 50 randomly extracted units.

| | IoU | ExplCov | SampleCov | ActCov | DetAcc | LabMask |
|---|---|---|---|---|---|---|
| NetDissect | $0.05 \pm 0.05$ | $0.32 \pm 0.24$ | $0.25 \pm 0.17$ | $0.23 \pm 0.20$ | $0.08 \pm 0.07$ | $0.28 \pm 0.17$ |
| CoEx | $0.09 \pm 0.05$ | $0.31 \pm 0.22$ | $0.22 \pm 0.15$ | $0.19 \pm 0.17$ | $0.09 \pm 0.07$ | $0.26 \pm 0.15$ |
| Our | $0.15 \pm 0.05$ | $0.68 \pm 0.31$ | $0.64 \pm 0.26$ | $0.39 \pm 0.18$ | $0.20 \pm 0.10$ | $0.55 \pm 0.17$ |
| Cluster 1 | $0.28 \pm 0.02$ | $0.99 \pm 0.00$ | $0.97 \pm 0.01$ | $0.56 \pm 0.02$ | $0.35 \pm 0.02$ | $0.63 \pm 0.08$ |
| Cluster 2 | $0.15 \pm 0.01$ | $0.99 \pm 0.01$ | $0.84 \pm 0.03$ | $0.57 \pm 0.04$ | $0.17 \pm 0.02$ | $0.68 \pm 0.08$ |
| Cluster 3 | $0.11 \pm 0.02$ | $0.61 \pm 0.27$ | $0.60 \pm 0.09$ | $0.28 \pm 0.10$ | $0.17 \pm 0.05$ | $0.59 \pm 0.10$ |
| Cluster 4 | $0.10 \pm 0.06$ | $0.42 \pm 0.15$ | $0.49 \pm 0.16$ | $0.27 \pm 0.09$ | $0.18 \pm 0.09$ | $0.53 \pm 0.15$ |
| Cluster 5 | $0.09 \pm 0.06$ | $0.40 \pm 0.21$ | $0.30 \pm 0.16$ | $0.28 \pm 0.20$ | $0.12 \pm 0.08$ | $0.34 \pm 0.15$ |

## B.3 ImageNet

Table 7 and Table 8 compare NetDissect, CoEx, and Clustered Compositional Explanations when the ResNet18 [17] and VGG-16 [41] are pretrained on the ImageNet dataset [9]. We can see that the differences among algorithms are similar to the ones discussed in Section 4.3, and thus, the insights are valid across different datasets.

Table 7: Avg. and Std Dev. of the desired qualities over the labels returned by NetDissect, CoEx, and Clustered Compositional Explanations (our) applied to ResNet pre-trained on ImageNet. Results are computed for 50 randomly extracted units.

|  | IoU | ExplCov | SampleCov | ActCov | DetAcc | LabMask |
|---|---|---|---|---|---|---|
| NetDissect | $0.04 \pm 0.02$ | $0.13 \pm 0.17$ | $0.36 \pm 0.27$ | $0.10 \pm 0.12$ | $0.12 \pm 0.11$ | $0.37 \pm 0.24$ |
| CoEx | $0.05 \pm 0.03$ | $0.13 \pm 0.16$ | $0.28 \pm 0.20$ | $0.12 \pm 0.08$ | $0.12 \pm 0.08$ | $0.32 \pm 0.19$ |
| Our | $0.13 \pm 0.08$ | $0.65 \pm 0.37$ | $0.65 \pm 0.29$ | $0.36 \pm 0.19$ | $0.18 \pm 0.11$ | $0.56 \pm 0.18$ |
| Cluster 1 | $0.28 \pm 0.02$ | $0.99 \pm 0.01$ | $0.98 \pm 0.02$ | $0.54 \pm 0.02$ | $0.36 \pm 0.03$ | $0.64 \pm 0.06$ |
| Cluster 2 | $0.15 \pm 0.03$ | $0.95 \pm 0.12$ | $0.89 \pm 0.06$ | $0.53 \pm 0.07$ | $0.18 \pm 0.04$ | $0.73 \pm 0.08$ |
| Cluster 3 | $0.13 \pm 0.05$ | $0.79 \pm 0.23$ | $0.66 \pm 0.11$ | $0.39 \pm 0.12$ | $0.10 \pm 0.03$ | $0.63 \pm 0.11$ |
| Cluster 4 | $0.07 \pm 0.03$ | $0.32 \pm 0.21$ | $0.42 \pm 0.18$ | $0.18 \pm 0.10$ | $0.12 \pm 0.06$ | $0.48 \pm 0.15$ |
| Cluster 5 | $0.06 \pm 0.04$ | $0.19 \pm 0.16$ | $0.29 \pm 0.19$ | $0.15 \pm 0.11$ | $0.12 \pm 0.08$ | $0.34 \pm 0.18$ |

Table 8: Avg. and Std Dev. of the desired qualities over the labels returned by NetDissect, CoEx, and Clustered Compositional Explanations (our) applied to VGG16 pre-trained on ImageNet. Results are computed for 50 randomly extracted units.

|  | IoU | ExplCov | SampleCov | ActCov | DetAcc | LabMask |
|---|---|---|---|---|---|---|
| NetDissect | $0.03 \pm 0.02$ | $0.13 \pm 0.18$ | $0.35 \pm 0.26$ | $0.11 \pm 0.11$ | $0.11 \pm 0.11$ | $0.39 \pm 0.25$ |
| CoEx | $0.04 \pm 0.03$ | $0.10 \pm 0.10$ | $0.28 \pm 0.19$ | $0.10 \pm 0.07$ | $0.10 \pm 0.09$ | $0.33 \pm 0.18$ |
| Our | $0.08 \pm 0.06$ | $0.38 \pm 0.34$ | $0.45 \pm 0.26$ | $0.22 \pm 0.17$ | $0.11 \pm 0.07$ | $0.36 \pm 0.16$ |
| Cluster 1 | $0.16 \pm 0.06$ | $0.92 \pm 0.16$ | $0.78 \pm 0.16$ | $0.48 \pm 0.13$ | $0.20 \pm 0.07$ | $0.38 \pm 0.13$ |
| Cluster 2 | $0.07 \pm 0.03$ | $0.43 \pm 0.24$ | $0.56 \pm 0.20$ | $0.20 \pm 0.10$ | $0.10 \pm 0.05$ | $0.43 \pm 0.17$ |
| Cluster 3 | $0.06 \pm 0.03$ | $0.29 \pm 0.21$ | $0.40 \pm 0.18$ | $0.18 \pm 0.11$ | $0.08 \pm 0.04$ | $0.39 \pm 0.15$ |
| Cluster 4 | $0.08 \pm 0.04$ | $0.16 \pm 0.15$ | $0.28 \pm 0.16$ | $0.13 \pm 0.10$ | $0.08 \pm 0.04$ | $0.33 \pm 0.15$ |
| Cluster 5 | $0.05 \pm 0.04$ | $0.10 \pm 0.12$ | $0.24 \pm 0.18$ | $0.12 \pm 0.10$ | $0.08 \pm 0.07$ | $0.28 \pm 0.17$ |

## C  Number of clusters

Table 9 shows the variation in explanations' quality when the Clustered Compositional Explanations algorithm uses a different number of clusters. We can observe a marginal loss in qualities when increasing the number of clusters. Moreover, by manually inspecting the returned labels, we found that several labels are repeated over the clusters, and less than ~30% of the labels are novel with respect to the usage of fewer clusters. We hypothesize that these results can open the door to further research on a novel algorithm that reduces repeated labels over clusters and finds the optimal number of clusters.

Table 9: Avg. and Std Dev. of the desired qualities over the labels returned by Clustered Compositional Explanations using a variable number of clusters. Results are computed for 50 randomly extracted units.

| Clusters | IoU | ExplCov | SampleCov | ActCov | DetAcc | LabMask |
|---|---|---|---|---|---|---|
| 5 | $0.15 \pm 0.37$ | $0.60 \pm 0.23$ | $0.70 \pm 0.17$ | $0.34 \pm 0.11$ | $0.21 \pm 0.10$ | $0.55 \pm 0.22$ |
| 10 | $0.09 \pm 0.05$ | $0.48 \pm 0.35$ | $0.68 \pm 0.23$ | $0.30 \pm 0.16$ | $0.13 \pm 0.06$ | $0.58 \pm 0.17$ |
| 15 | $0.07 \pm 0.04$ | $0.44 \pm 0.36$ | $0.66 \pm 0.21$ | $0.25 \pm 0.16$ | $0.09 \pm 0.05$ | $0.56 \pm 0.18$ |

## D  Other Statistics

This section lists additional statistics or metrics that can be used with our identified qualities to inspect models and explanations.

**Scene Percentage**  Percentage of scene concepts associated with explanations. Scene concepts are concepts whose masks cover the full input. Harth [16] argues that a high percentage of scene concepts is undesirable in some circumstances.

$$SccnePerc(\mathfrak{D}, \mathfrak{L}^{best}) = \frac{\sum_{L \in \mathfrak{L}^{best}} |\{t : t \in L \wedge (|S(x,t)| = n_s \forall x \in \mathfrak{D})\}|}{\sum_{L \in \mathfrak{L}^{best}} |\{t : t \in L\}|} \tag{44}$$

where $\mathfrak{L}^{best}$ is the set that includes the labels associated with each neuron by the algorithm.

**ImRoU**  It is a modified version of the IoU score where sparse overlapping between concepts' masks and activations are penalized. It has been used as an optimization metric to reduce the number of scene concepts [16]. It is based on the idea of penalizing concepts whose activation is distributed like in a random activation.

$$ImRoU_r(\mathfrak{D}, [\tau_1, \tau_2], L) = \frac{\sum_{x \in \mathfrak{D}} |M_{[\tau_1, \tau_2]}(x) \cap S(x, L)| - r \times \frac{1}{n_s} \times |M_{[\tau_1, \tau_2]}| \times |S(x, L)|}{|M_{[\tau_1, \tau_2]}(x) \cup S(x, L)|} \tag{45}$$

where the constant $r$ controls the weight of the random intersection.

**Pearson Coefficent**  It measures the correlation between the IoU score, the firing rate of a neuron, and the performance. This metric has been used to measure the correlation between the interpretability of a latent space and performance. It is not directly connected to the quality of the returned explanations.

$$Pearson = PearsCoeff(IoU_{\mathfrak{X}}, Accuracy_{\mathfrak{X}}) \tag{46}$$

where $\mathfrak{X}$ is the set of samples where the neuron fires in the considered range $[\tau_1, \tau_2]$, and IoU is the set of intersections over union per sample.

**Average Activation Size**  The average size of the masks covering the considered range $[\tau_1, \tau_2]$ over the dataset.

$$AvgActSize(\tau_1, \tau_2, \mathfrak{D}) = \frac{\sum_{x \in \mathfrak{D}} |M_{[\tau_1, \tau_2]}(x)|}{\sum_{x \in \mathfrak{D}} n_s} \tag{47}$$

**Average Label's Mask Size**    The average size of the label's masks over the dataset.

$$AvgLabSize(L, \mathfrak{D}) = \frac{\sum_{x \in \mathfrak{D}} |S(x, L)|}{\sum_{x \in \mathfrak{D}} n_s} \tag{48}$$

**Average Overlapping**    The average size of the overlapping between label's masks and neuron's activation covering the considered range $[\tau_1, \tau_2]$ over the dataset.

$$AvgOverlap(L, \tau_1, \tau_2, \mathfrak{D}) = \frac{\sum_{x \in \mathfrak{D}} |M_{[\tau_1, \tau_2]}(x) \cup S(x, L)|}{\sum_{x \in \mathfrak{D}} n_s} \tag{49}$$

**Absolute Label Masking**    The difference in the neuron's activations between normal and fully masked inputs where only the associated label is kept. A low score means the linkage between the label and the activation range is strong. With respect to the Label Masking presented in Section 3.2, this variant is not normalized, and thus, it is difficult to compare different neurons or aggregate their scores. In preliminary experiments, we observe that the mean scores reached in this metric are similar among different algorithms, and they suffer from high variance due to the different ranges captured by different neurons.

$$LabMask(L, \tau_1, \tau_2, \mathfrak{D}) = \sum_{x \in \mathfrak{D}} |(A_{[\tau_1, \tau_2]}(\theta(x, L)) - \sum_{x \in \mathfrak{D}|} |A_{[\tau_1, \tau_2]}(x) \cap S(x, L)| \tag{50}$$

# E    Activation Importance

Table 10 shows how many times the network changes its prediction when we mask the activations covered by each cluster or covered by the CoEx thresholds. Intuitively, if an activation band is not used in the decision process, then it should never change the prediction if it is masked out.

Conversely, we can observe that the change in prediction is similar in almost all the clusters but Cluster 1, which often contains default rules and unspecialized activations. This means that the full spectrum of activations has an impact on the decision process.

Table 10: Percentage of predictions changed when the activations of a given cluster are masked. In the case of Compositional Explanation (CoEx), we mask all the activations above the 0.005 percentile.

|  | Prediction Change % |
| --- | --- |
| CoEx | 12.9 |
| Cluster 1 | 9.7 |
| Cluster 2 | 12.90 |
| Cluster 3 | 12.29 |
| Cluster 4 | 11.62 |
| Cluster 5 | 13.79 |

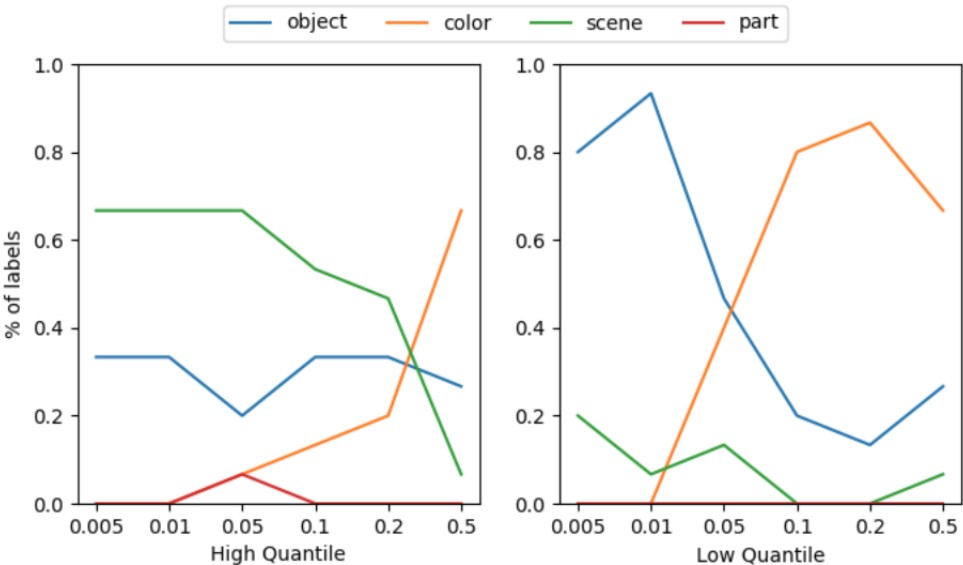

Figure 6: Percentage of labels' category associated with the labels returned by CoEx over different activation ranges. Left: ranges from quantile to infinite. Right: range from 0 to quantile. Results are computed over 100 randomly extracted units.

## F    Theshold Impact

In this section, we analyze the impact of the threshold's value. Specifically, we analyze the category distribution of the labels returned by the NetDissect and Compositional Explanations when the threshold is lowered. We first consider the following list of top quantiles ranges: $\{[0.005, \infty], [0.01, \infty], [0.05, \infty], [0.1, \infty], [0.2, \infty], [0.5, \infty]\}$ and then we apply similar ranges but extracting the lowest quantiles, namely $\{[\epsilon, 0.005], [\epsilon, 0.01], [\epsilon, 0.05], [\epsilon, 0.1], [\epsilon, 0.2], [\epsilon, 0.5]\}$ where $\epsilon = 1e - 6$.

Figure 6 shows the percentage of the labels associated by CoEx to 100 randomly extracted units when changing the threshold. We can observe that lowering the threshold (or equivalently increasing the range of considered quantiles) penalizes some label categories and rewards others. For example, we can observe that the colors benefit from larger ranges while objects benefit from smaller ones.
At first sight, one could hypothesize that this behavior is due to the larger size of the mask $M_{[\tau_1, \tau_2](x)}$ generated by a lower threshold since a larger mask increases the intersection of a large concept when this is not fully detected using a higher threshold. However, this observation is not enough to explain the results. Indeed, lowering the threshold also increases the intersection of small concepts. Since their $IoU$ scores converge to 1 faster than ones of large concepts, the distribution of returned labels should be similar on average. reports the average segmentation area per category, confirming this analysis. We can observe that two categories with similar areas, like Color and Object, have opposite behavior in the plots of Figure 6. Moreover, we can see that the distribution of categories is not consistent when considering high and low quantiles, even though their range size is equal.
Putting all together, we explain these results using the main hypothesis of this paper: neurons recognize different concepts at different *activation levels*.

## G    Additional Details about the Setup

All the models considered in this paper have been pre-trained on the Place365 dataset [45]. In particular, the checkpoints used are the same used in the CoEx and NetDissect papers. Annotation for Pascal [12] and Ade20K [44] datasets are retrieved from the Broden dataset [2].

Regarding the formulas' length, we fix the limit to 3, as previously done in literature by Makinwa et al. [27], Harth [16], Massidda and Bacciu [28]. According to the results of Mu and Andreas [30], increasing the formula's length should not impact the results presented in this paper. Another

difference with respect to the implementation of CoEx is that we actively check logical equivalences between formulas. This difference means that we use a beam of size 10 only during the first beam, and then we set the beam size to 5 to replicate the configuration of Mu and Andreas [30].

Finally, we choose to use a clustering algorithm over manually splitting the activation space since we desire clusters that aggregate activations associated with a shared semantic (e.g., all the activations that recognize a car inside a single cluster). Conversely, a manual split (e.g., using the percentile) can often separate activations associated with the same concept/s to multiple subsets. In this case, the concept can be overlooked by the algorithm since the overlapping mask is split into different subsets, or multiple splits could be associated with the same label, thus penalizing other concepts.

