# OpenReview forum: "Towards a fuller understanding of neurons with Clustered Compositional Explanations"
_NeurIPS.cc/2023/Conference — NeurIPS 2023 poster_

### Official Review · Reviewer_4jti · 2023-06-13

**Soundness:** 2 fair
**Presentation:** 2 fair
**Contribution:** 1 poor
**Rating:** 2
**Confidence:** 5

**Summary:**

This paper is a niche extension of seminal work on network dissection. The authors present a generalization, called Clustered Compositional Explanations, that combines Compositional Explanations with clustering and a novel search heuristic to approximate a broader spectrum of the neuron behavior.

**Strengths:**

Easy to read

**Weaknesses:**

cf. Limitations

**Questions:**

N/A

**Limitations:**

This is mainly a search-based approach on a set of clusters which group neurons.
This is very incremental work. Research angle of the work is limited to the application of search on clusters grouping neurons
No novelty

---

> ### Author Rebuttal · Authors · 2023-08-09
>
> Thank you for taking the time to review our paper.  We understand that our work did not present well to the reviewer, but perhaps that is due to a misunderstanding of the paper. First of all, the reviewer mentions the research of a “set of clusters which group neurons.” However, we never grouped neurons in our paper.  Instead, clusters group activations of a single neuron over the full dataset, and this is a big and crucial difference that maybe can denote a superficial reading. The topic of the paper is the analysis of the full spectrum of the neurons' behavior, and thus it is inside the scope of the venue. Moreover, the original technique to compute compositional explanations (the paper that we generalize) is search-based, and it was chosen as an oral presentation for this same conference (NeurIPS). Therefore, we think that the usage of search-based techniques is not a limitation for the current venue; rather it is a trend of high-quality work.
>
> We are grateful for your comments and respect your concerns about the novelty.  However, to our knowledge, this is the first work to cluster and analyze different ranges of activations in terms of compositional explanations (or dissection).  Further, it is the first work that proposes heuristics to speed up the computation of compositional explanations.

---

### Official Review · Reviewer_2vbF · 2023-07-04

**Soundness:** 3 good
**Presentation:** 4 excellent
**Contribution:** 2 fair
**Rating:** 6
**Confidence:** 4

**Summary:**

The authors propose a novel XAI method called Clustered Compositional Explanations (CCE), that aims to descibe the function that a group of neurons in a neural network perform. The method is built on top of CoEx (Mu and Andreas 2020) and NetDissect (Bau et al 2017) with the novelty being its generalization to multiple activation ranges instead of high activation thresholds only. The objective for CCE aims to maximize the IoU score between the activations of an input and a neuron in the range of min and maximum activation thretholds across all clusters, over all set of logical connections between the annotated sets of concepts in the dataset. The authors solve the optimization problem by presenting a heuristic algorithm MMESH. Results from the authors indicate that the explanations obtained from proposed method does better than NetDissect and CoEx on various metrics that include accuracy and coverage among others.

**Strengths:**

The connection between Clustered Compositional Explanations and existing neuronal explainability methods CoEx and NetDissect is explained very well. The proposed method generalizes both these methods, and the reviewer agrees with the authors that explaining groups of neurons over larger activation ranges could lead to broader explanations. This is confirmed in the results, as the proposed CCE method outperforms CoEx and NetDissect on various metrics, but especially so on coverage related metrics.

**Weaknesses:**

1. My biggest concern is with the MMESH algorithm, and while it is admissible as shown in the appendix, the algorithm is tailored to the CCE method and not easy to follow. Unless the authors plan to release their code (not included in submission), I believe it will greatly limit the utility of their approach since their is a higher barrier of implementation for practitioners versus CoEx.

2. I would like to see results across a broader range of models and datasets in order to ensure that the proposed method does not overfit to explaining the neurons of a particular model, dataset, and training procedure. I think this is reasonable since MMESH takes less than 2 minutes per unit (L219). All results have been reported on CNNs (AlexNet, ResNet, DenseNet) which I presume are trained with supervised learning.

**Questions:**

1. Could the authors please explain why having more clusters than one is necessary, since in Table 2 the first cluster has perfect dataset and sample coverage and outperforms the averaged across clusters on all desiderata, similarly for Table1, 3 in the appendix.

**Limitations:**

The authors have both mentioned potential limitations and highlighted avenues for future work. See also Weaknesses.

---

> ### Author Rebuttal · Authors · 2023-08-09
>
> We thank the reviewer for the appreciation of the important research questions addressed by our paper and our presentation. We hereby clarify and answer point by point both the questions **(Q)** and highlighted weaknesses **(W)**.
>
> **W1)** We can assure the reviewer that, as written in the checklist, the code will be released upon acceptance. We greatly believe that releasing the code is a fundamental step for research in Deep Learning. Regarding the MMESH heuristic, it is tailored to the Compositional Explanation algorithm, not to the Clustered variant. It can be used (and it is used in our code) to also speed up the vanilla CoEx algorithm. An ablated version of it (Areas Heuristic) can also be used to speed up Network Dissection, as noted in Appendix  B.2.
>
> **W2)**  As written in Section 4.1, we follow the same setup of Mu and Andreas [24]. Therefore, all the models are trained on the Place365 dataset, and we use the publicly available models used by both Compositional Explanations and Network Dissection. They are the standard models commonly used by all the literature on the topic (see Section 2). The procedure has been motivated in literature by the fact that the concepts annotated in the Ade20k dataset are connected to the Place365 dataset.
>
> To address the concern of the reviewer, we added new experiments in the file attached to the rebuttal, where we show (Table 2 and Table 3) that the results hold even when models are trained on ImageNet, despite the fact the concepts are weakly connected to the trained model. In this case, we tested ResNet and VGG16 (which is an additional model to the pool of already tested ones). We can observe that the results are similar to the ones obtained before. Moreover, in Appendix C2, we tested the method using a different concept dataset. Thus, given the new and old results, the methods do not depend on any particular used training procedure, architecture, or concept dataset.
>
> **Q1)** The short answer is that all the clusters have an impact on the decision process (Table 1 of the pdf attached to the rebuttal) and that the goal of the paper is to provide a broader view of the neuron behavior by extracting as many concepts recognized by the neuron as possible.
>
> In more details, there are several reasons to consider multiple clusters instead of a single one. Note that: the long-term goal is to understand exactly which (and all) the concepts the neuron recognizes; and that a feature map (i.e., the output of a neuron in our case) is generally composed of multiple activations associated with multiple clusters.
>
> If we consider a single cluster that includes all the possible activations, then the algorithm will assign to that single cluster the n concepts (in our case n=3) that overlap the most. Since a single cluster that covers all the activation means that there is no mask applied to the activations, then the algorithm simply will select the concepts whose annotations are the biggest ones (i.e., it is a degenerate case), and thus the explanations do not depend on the given model/neuron.
>
> Now, let us consider the case where we compute multiple clusters and consider as an explanation only the one associated with the cluster with the best scores (i.e., usually Cluster 1 as you noted). In this case, we are providing as an explanation the concepts recognized by the lowest activations (e.g., person). Therefore, if the neuron recognizes other concepts (e.g., ice cream) using a different activation band (e.g., the highest ones as in NetDissect) we are not including them in our explanation since we only selected the one of Cluster 1. And this is in contrast with the goal of our paper, which is extracting a broader view of the concepts the neuron is able to recognize with respect to CoEx and NetDissect, which can be seen as methods that use a single cluster. In an ideal scenario, as written in the conclusion, we would like to obtain a clustering algorithm that assigns a cluster for each set of semantically connected concepts recognized by the neuron, and thus obtaining complete explanations. This paper represents a step in this direction.
>
> Finally, in Table 1 of the pdf attached to the rebuttal, the reviewer can observe that almost all the cluster have the same impact on the decision process, and thus, considering only one cluster ignore part of the decision process.

---

> > ### Comment · Reviewer_2vbF · 2023-08-14
> > **Response to rebuttal**
> >
> > Thank you for taking the time to write a thorough rebuttal, as well as for adding the experiments related to testing explainability on ImageNet. I think it makes the work more comprehensive, and the authors' release of the CoEx code should make it easier for others to use their method as well. I have therefore revised my rating from a Borderline Accept to a Weak Accept.

---

### Official Review · Reviewer_CMC6 · 2023-07-04

**Soundness:** 3 good
**Presentation:** 3 good
**Contribution:** 3 good
**Rating:** 6
**Confidence:** 5

**Summary:**

This paper extends the ideas of Mu (2020) to examine a more powerful class of compositional explanations of neurons, by adding the goal of explaining other ranges of neuron activations, unlike previous work that had restricted analysis to the top ranges only.  Like the previous work by Mu, the paper searches for compositions of human-understandable concepts that explain a neuron’s behavior, but the proposed method begins by first subdividing neuron activations into ranges. Adding the ability to explain lower ranges results in much better coverage: it allows the authors to create explanations of a larger portion of neurons’ behavior, and as a result this paper achieves much higher coverage metrics, explaining more neurons with high matching scores, compared to previous methods.


**Strengths:**

The main strength of this paper is the way that it brings a systematic interpretability analysis to neurons and ranges of neuron activations that have not previously been systematically analyzed.

The paper finds that, at lower activation ranges, consistent interpretations can be found for most neurons. On the other hand, the paper finds that most of these interpretations are generic concepts like sets of colors, and the paper hypothesizes that these observed labels are “default labels” that describe inputs when neurons are in a random state. The paper applies their explanations to neurons in a random untrained network to validate this hypothesis; that is a good baseline-setting experiment.

The paper also finds that as neuron activations rise to higher levels, the explanations of their behavior becomes progressively more specific.  Unsurprisingly, when analyzing the entire activation range of neurons, the paper finds that more individual neurons activate on several unrelated concepts over their range, observing a higher level of such polysemanticity than previous works, but interestingly, the paper finds a portion (15%) of neurons that have a consistent semantics over their range.


**Weaknesses:**

On weaknesses: beyond the clever uninitialized-network experiment, the paper does not conduct experiments that would triangulate the proposal that the intermediate-activation states of neurons might be meaningful.  For example, it is natural to ask: if there are ranges of concepts that seem to match middle-range activations, is the network actually "looking" for those midrange concepts?  For example, will forcing those neurons to those middle ranges cause a network to change its predictions towards classes that correspond to those explained mid-range concepts?

The paper makes a good contribution by increasing the observational breadth of neuron interpretations to include middle-range activations, but it does not directly defend the idea that such midrange activations will be useful to understand.

**Questions:**

The main question is the one posed above.  How can we know that the concepts matched in middle ranges of activations are meaningful to map out?

Specifically: the paper observes that that the neurons sometimes match different concepts in the middle ranges than they do at extreme ranges.  Do the mid-range matched concept correspond to meaningful decisions by the network?  For example, would forcing neurons to those middle ranges cause the network to behave as if those middle-range concepts are detected? Does it cause the network to behave as if different concepts were detected compared to those same neurons set to the highest ranges or lowest ranges?

**Limitations:**

The authors note several of the potential limitations of their methods and address some of them in the appendix.

No immediate negative societal impacts are anticipated from this line of work.

---

> ### Author Rebuttal · Authors · 2023-08-09
>
> We thank the reviewer for the meaningful suggestions. Following the recommendation, we ran an experiment to better consider middle activations, and we feel that the addition of this experiment to the appendix makes the paper stronger.
> In particular, we tested how many times the network changes its prediction when we mask the activations covered by each cluster or covered by the CoEx thresholds (see Table 1 in the pdf attached to the rebuttal). Intuitively, if an activation band is not used in the decision process, then it should never change the prediction if it is masked out.
>
> Conversely, we can observe that the change in prediction is similar to the CoEx ranges in almost all the clusters but Cluster 1, which often contains default rules and unspecialized activations, as described in the paper. This means that the full spectrum of activations has an impact on the decision process, as we hypothesized at the beginning of our paper. We prefer this experiment over imposing a middle range over all the activations since the ranges are usually tailored to specific positions in the input (as shown in Figure 4 in the main paper), and it is difficult to impose a realistic range (in distribution) over different positions.

---

> > ### Comment · Reviewer_CMC6 · 2023-08-20
> >
> > Thank you for the added measurements of prediction changes across ranges.

---

### Official Review · Reviewer_rkdS · 2023-07-22

**Soundness:** 2 fair
**Presentation:** 2 fair
**Contribution:** 2 fair
**Rating:** 6
**Confidence:** 3

**Summary:**

This paper focuses on a problem with Network Dissection and Compositional Explanation methods: these two methods explain the concept encoded by a neuron (or, more precisely, a convolutional filter) by only considering highly activated regions in the feature map. To address the problem, this paper proposes to divide the activation values into different ranges (clusters) and generate an explanation for each range of activation value. This paper also proposes a heuristic to accelerate the search for the optimal logical combinations of concepts in the Compositional Explanation method.

**Strengths:**

1. This paper focuses on an important issue, i.e., previous concept-based explanation methods, such as Network Dissection or Compositional Explanation, neglect feature map regions with low activation values.
2. The authors theoretically prove that the proposed heuristic is admissible, which guarantees the optimality of the solution found using the heuristic.


**Weaknesses:**

1. The contributions are incremental and miscellaneous. From my view, the contribution of this paper is mainly two-fold. The first contribution is that the paper considers different ranges of activation values to address the problem with previous explanations (Network Dissection and Compositional Explanation) that they only analyze the feature map regions with extremely high activation values. However, the proposed method is only a simple extension of previous methods, and the motivation of using clustering instead of manually dividing the activation values into different ranges is not well explained. The second contribution is to propose a heuristic searching method to accelerate the Compositional Explanation method. However, the second contribution is not inherently correlated with the first one, and I feel the paper is separated into two uncorrelated parts.

2. Many of the notations are inconsistent or confusing. For example:
(a)	In Line 106, the intersection size is defined as $IMS(x,L)$. But in Eq. (5) to Eq. (7), it is denoted as $IMS_{[\tau\_1, \tau\_2]}(x,L)$. The meaning of the subscripts needs to be clarified.
(b)	In Eq. (1), $\mathcal{L}^n$ should be $L$, in order to be consistent with Eq. (2).
(c)	In Eq. (5)-(7), there is a notation $L_L$. I’m confused about what the two L mean respectively. If they refer to different meanings, please use different letters.
(d)	In Eq. (5)-(7), there is no explanation for the subscript $R$ in the notation $L_R$.
(e)	In many equations, such as Eq. (2), Eq. (11), Eq. (13), Eq. (14), an equal sign is missing.

3.	The metric Activation Coverage seems redundant. Given the metric Intersection Over Union and the metric Detection Accuracy, we can actually derive the metric Activation Coverage by using the inclusion-exclusion principle $|A\cup B|=|A|+|B|-|A\cap B|$. Therefore, I doubt this metric is redundant.


**Questions:**

1. In Table 2, when computing the desiderata qualities (or metric scores) of the proposed method, the authors simply average the scores from the 5 clusters. However, since different clusters correspond to different ranges of activation values, does it make more sense to assign different weights to the scores of these clusters? For example, it is natural to assign higher weights to clusters with higher activation values. Nevertheless, the authors are also encouraged to explain why doing simple averaging is a reasonable choice.

2. As noted by the authors in Line 259 to Line 276, the concept labels obtained from Cluster 0 and Cluster 1 usually represent the “default labels” to which the algorithm converges when the activations are random. This implies that the labels obtained from Cluster 0 and Cluster 1 actually cannot explain what the model has learned from the data. In this way, the metric scores of Cluster 0 and Cluster 1 are meaningless and should not be used to compute the final score of the proposed method.

3. Why do the authors set the number of clusters to 5? If there are more or fewer clusters, will the analysis and conclusion in Sections 4.3 and 4.4 change? I suggest the authors conduct ablation studies regarding the number of clusters and clearly explain how it will affect the empirical results.

4. In Table 3, why do the results with ReLU greatly differ from results without ReLU? Could the authors give some explanations or discussions on this issue?



**Limitations:**

Yes, the authors have discussed the limitations and broader impact of their work.

---

> ### Author Rebuttal · Authors · 2023-08-09
>
> We thank the reviewer for the appreciation of the important research questions addressed by our paper. We hereby clarify and answer point by point both the questions **(Q)** and highlighted weaknesses **(W)**.
>
> **W1)** While we do agree that our paper is an extension of previous work, we stress the fact that every generalization paper is, by definition, an extension. And the compositional explanation paper is an extension of NetDissect too. We believe that Deep Learning and XAI needs work that generalizes previous claims to broader contexts.
>
> Regarding the choice of clustering over manual splitting the activation space, we will add a discussion about it in the appendix.  Briefly, the choice is motivated by wanting clusters that aggregate activations associated with a common semantic. By doing a manual split (e.g., using the percentile)  it is more likely to separate activations that are associated with the same concept/s. Using a clustering algorithm mitigates this problem.
> As noted in the conclusion section, this opens a promising future research direction: the development of a clustering algorithm tailored to the task to solve the problem completely.
>
> Finally, regarding the two contributions, note that the MMESH heuristic is a practical pre-requisite for the clustered compositional explanation in order to make their computation feasible in a reasonable time. While it could be possible to propose MMESH without Clustered Compositional Explanations, the inverse does not hold, and thus, in our opinion, the two contributions are chained and linked.
>
> **W2)** Thank you for pointing out some incontinence in the notation. We think these inconsistencies are easy to fix. We will follow your suggestions for points a), b), and e). Regarding c) and d) $L_L$ and $L_R$ are the formula's left side and right side connected by the “op” operator. And we use L since both sides are labels. We will fix them by better specifying the subscript notation in the camera-ready version and we are considering replacing L and R with left and right arrow symbols.
>
> **W3)** It is true that the denominators of the three metrics are connected, and thus there is a dependency between them.  However, the metrics refer to and express different qualities of the compositional explanations. Moreover, it is not possible to obtain Activation Coverage from the mere scores of Detection Accuracy and IoU; you need the quantities needed to compute them. For these reasons, we keep Activation Coverage as a separate metric.
>
> **Q1)** As highlighted in the paper, our opinion is that the best way to compare different methods is to look at the scores of each cluster separately since one or two big clusters can influence the average scores.
> Regarding the weights, finding the best way to assign them is not trivial, and it is probably worth its own proper investigation. Each of the basic weighting mechanisms (e.g., based on the size of the cluster or the impact of it) has its own caveat. For example, assigning high scores to high activations is debatable since the findings of our paper. Table 1 in the rebuttal file shows that middle activations have the same impact on the decision process of high activations. Moreover, while high activations are rare, low activations are common, and one could argue that an explanation should capture the most common behavior. We thank the reviewer for the interesting idea, which we will add as a future direction.
>
> **Q2)** As written in the previous answer, our opinion is that the best way is looking at the scores of each cluster separately instead of trying to remove the terms from the average. By analyzing multiple metrics and multiple clusters, one could obtain a better idea of the quality of explanations deeper understanding of the neurons' behavior and the quality of the returned explanations instead of analyzing the average of one metric. Regarding the possibility of ignoring lower clusters, there are cases (e.g., the last layer of DenseNet, as shown in the paper) where Cluster 0 and Cluster 1 include specialized activations. Moreover, even when Cluster 0 and Cluster 1 are unspecialized on average, there are units that could have specialized lower activations, and thus, they should be taken into account in the average scores. Therefore, we think that removing entire clusters from the average is not the right approach.
>
> **Q3)** We performed this study in Appendix D. We observed a marginal loss in qualities when increasing the number of clusters. Moreover, we found that several labels are repeated over the clusters, and less than ~30% of the labels are novel with respect to the usage of fewer clusters. Since there is no gain in increasing the number of clusters and it is more difficult to evaluate a greater number of explanations, we fixed the number of clusters to 5. However, as previously written, we think that a promising direction for the future is to develop a clustering algorithm tailored to the task, which could find the optimal clusters (in terms of number and semantics).
>
> **Q4)** We think that the behavior is not an issue but just a different way in which different layers use the activations and parse the concepts. And actually, despite the differences, we can find similarities in the behavior of different layers when activations are close to 0. In ReLU layers, activations are stored in Cluster 1, and they are unspecialized 93% of the time. This percentage becomes smaller when we go up towards the higher clusters. We can observe similar behavior in the case of the layer without ReLU reported in Table 3 of the main paper. Here, since the activations can assume negative values, the activations close to zero are stored in the middle clusters. And indeed, we can observe that Cluster 3 includes unspecialized activations 95% of the time. And again, when we move far away from zero, the percentage starts to decrease, as in the ReLU layers. We will add this discussion to the final paper.

---

> > ### Comment · Reviewer_rkdS · 2023-08-16
> >
> > Since my major concerns are addressed by the authors' rebuttal, I would like to raise my rating to 6.

---

### Official Review · Reviewer_BxbC · 2023-07-24

**Soundness:** 3 good
**Presentation:** 2 fair
**Contribution:** 3 good
**Rating:** 6
**Confidence:** 2

**Summary:**

The paper represents a generalization of compositional explanation called clustered compositional explanations which combines compositional explanations with clustering and a search heuristic to approximate a broader spectrum of the neuron behavior, by proposing the Min-Max Extension per Sample Heuristic (MMESH). This paper gives an analysis of the phenomena connected to the neuron's activations like the unspecialization of the lowest activations in ReLU networks and the progressive specialization.

**Strengths:**

1. This paper well delivers its contribution on the generalization of CoEx based on a heuristic and a wider spectrum of activations by clustering them.
3. The experiments are thoroughly conducted with a detailed analysis of the proposed MMESH. The paper also addresses limitations with future direction of the research.

**Weaknesses:**

1. It would be nicer to clearly show the difference between Mu and Andreas [24].

**Questions:**

Please refer to the weakness section.

**Limitations:**

Please refer to the weakness section.

---

> ### Author Rebuttal · Authors · 2023-08-09
>
> We thank the reviewer for the appreciation of our paper and our contribution.
>
> Regarding the difference between Mu and Andreas [24], given the additional page available in the camera-ready version, we plan to move Appendix A into the main paper to address the reviewer's concern and to better highlight mathematically the differences between our approach and theirs. The differences are in the algorithm: they use an exhaustive search while we propose a heuristic search, in the usage of clustering to compute thresholds instead of using a fixed ad-hoc threshold - used to mask the activation maps, and in the consideration of multiple intervals of activations and thus the full spectrum of the neuron's activations.
>
> Indeed, [24] uses a single threshold computed as 0.005 quantiles of the activation for each neuron and considers only activation above this threshold. Conversely, we compute multiple thresholds $\tau_1$, $\tau_2$..$\tau_n$ using clustering, and consider multiple intervals for each neuron ($[\tau_1,\tau_2], [\tau_1 \tau_2,..],[\tau_{n-1},\tau_n]$). In this way, we can analyze the full spectrum of activations for each neuron. Hopefully, adding Appendix A and this discussion to the paper will address the concern of the reviewer.

---

### Author Rebuttal · Authors · 2023-08-09

Dear reviewers,
we report the additional experiments requested by “Reviewer 2vbF” and "Reviewer CMC6”  in the file attached to this global comment. We validated the importance of the middle activation (**Table 1**) and tested our algorithm on models (VGG16 and ResNet18) trained on a different dataset and using a different training procedure (ImageNet) (**Table 2** and **Table 3**).
The new results confirm the broad scope of our findings and the importance of including all the activations in the explanation process.  We thank all of you for your suggestion since we feel that the requested fixes, the discussion, and these new results strengthen the paper.

---

### Decision · Program_Chairs · 2023-09-21

**Decision:**

Accept (poster)

**Comment:**

The paper proposes a method of compositional explanation of neurons where it considers different ranges of activations as opposed to just looking at high activations. The work serves as a promising extension to the previous works by targeting their limitations. I would like to see ablation experiments reported in the rebuttal as part of the paper and the additional experiments conducted to show the generalization of the approach.